# Biodegradable (PLGA) Implants in Pediatric Trauma: A Brief Review

**DOI:** 10.3390/children13010019

**Published:** 2025-12-22

**Authors:** Herman Nudelman, Tibor Molnár, Gergő Józsa

**Affiliations:** 1Department of Paediatrics, Clinical Complex, Division of Surgery, Traumatology and Otorhinolaryngology, University of Pécs, 7 József Attila Street, 7623 Pécs, Hungary; 2Department of Thermophysiology, Institute for Translational Medicine, Medical School, University of Pécs, 12 Szigeti Street, 7624 Pécs, Hungary

**Keywords:** PLGA, fracture, osteosynthesis, biodegradable, absorbable, poly-l-lactic-co-glycolic-acid

## Abstract

**Highlights:**

**What are the main findings?**

**What are the implications of the main findings?**

**Abstract:**

**Background/Objectives:** Biodegradable implants have emerged as a promising alternative to traditional metallic fixation devices in pediatric orthopedic surgery. Avoiding implant removal is especially advantageous in children, who would otherwise require a second operation with additional anesthetic and surgical risks. This study reviews the current use of poly(lactic-co-glycolic acid) (PLGA) implants in pediatric fracture fixation and evaluates how they address limitations associated with traditional hardware. **Methods:** A narrative review was conducted summarizing current evidence, clinical experience, and case examples involving PLGA-based devices used in pediatric trauma. Special emphasis was placed on the degradation mechanism of PLGA, its controlled hydrolysis profile, and the capacity of the material to provide temporary mechanical stability during bone healing before complete resorption. The review included studies of PLGA use in forearm, distal radius, ankle, and elbow fractures, comparing outcomes to those obtained with metallic implants. **Results:** Across multiple clinical reports and case series, PLGA implants demonstrated effective fracture healing, stable fixation, and complication rates comparable to traditional metallic devices. Patients treated with resorbable implants benefited from reduced postoperative morbidity, no requirement for implant removal, and improved imaging compatibility. **Conclusions:** PLGA-based bioabsorbable implants represent a safe and effective alternative to conventional metal fixation in children. Their favorable degradation kinetics and clinical performance support their growing use in pediatric trauma surgery, while ongoing advances in polymer design and bioresorbable alloys continue to expand future applications.

## 1. Introduction

Fractures in children are common, yet management still poses distinct challenges due to the growing skeleton [1,2]. Simple fractures may heal with casting alone but more complex injuries will require operative fixation [2]. Traditional implants made of stainless steel or titanium reliably stabilize fractures [3]. Still, in children, these permanent implants are to be removed after healing to prevent interference with bone maturation and to avoid long-term complications. The mandated double operation carries risks of infection, complications of anesthesia, and damage to tissues. It also adds psychological and economic burdens by exposing children and families to additional hospital visits and procedures [4]. Among other concerns, these mainly drove the interest in biodegradable implants, which can fix and hold fractures during healing and safely dissolve, obviating the need for removal [5,6]. Over the past few decades, advances in materials science have yielded bioresorbable polymers and alloys suitable for orthopedic use in both children and adults [6,7,8]. In this review, we focus on the clinical applications of biodegradable implants—particularly PLGA-based devices—in pediatric trauma, and we discuss their benefits, limitations, and prospects in improving fracture management for young patients.

Children’s bones differ from adults’ in structure and healing capacity. Pediatric long bones have growth plates (physes) and a thick periosteum, allowing many fractures to remodel with time [9,10]. However, surgical stabilization for accurate alignment and favorable outcome is needed in the case of open fractures, multi-fragmentary breaks, accompanying soft tissue damage, or fractures affecting the articular surfaces [11,12,13]. Metal implants are associated with reliable and durable support and a limited presence of foreign body reactions. Despite their effectiveness, removal is necessary to mitigate risks associated with growth disturbance or implant migration. Hardware traversing through or near a growth plate can cause growth arrest, and protruding or retained implants may irritate surrounding soft tissues, cause foreign body reactions, or lead to stress shielding and refractures [14]. Moreover, the presence of a stiff metal implant can potentially alter bone remodeling during growth. As mentioned, a removal procedure entails a repeat anesthesia and surgical dissection, which carries a risk of infection, nerve injury, and scarring. Particularly in small children, anesthesia exposure itself is not without concern. From a health system perspective, scheduled implant removals increase the overall cost and workload in trauma care. Consequently, there has been a long-standing motivation to develop fixation methods that render hardware removal obsolete while still ensuring a reliable fixation [14,15].

Synthetic copolymers are the most widely adopted bioresorbable appliances [6,16]. Poly(lactic-co-glycolic acid) (PLGA), a copolymer of lactic and glycolic acids, is a second-generation bioresorbable material with enhanced mechanical properties and highly predictable degradation kinetics [6]. Polymer-based implants break down through hydrolysis and secondary enzymatic reactions [17,18]. During degradation, lactic and glycolic acid monomers are produced, which form water and carbon dioxide through the Krebs cycle [19]. The degradation rate can be tailored for needs, making PLGA desirable for clinical settings [14,19]. Degradation is fastened by a higher glycolide content, while a greater proportion of lactide extends the implant’s lifespan [6,19]. In practice, PLGA constructs with an 85L:15G composition are reported to be mostly absorbed in about 12–16 months [19,20,21]. By two years post-injury, PLGA devices have largely dissolved, leaving behind only remodeled bone with no foreign material [22]. Mechanical performance of the implants could be a concern, since polymers are not as strong or stiff as metal. Even though it is a tough polymer, it possesses a lower bending modulus and tensile strength compared to stainless steel or titanium [23,24]. In practice, current resorbable implants such as PLGA pins and screws are made in sizes suitable for small bones and are intended for fractures where extreme forces are not expected before healing. Their ability to maintain fixation stability for the necessary duration has been supported by clinical studies, as discussed in the literature. During degradation, these implants typically lose strength gradually, which is ideal: the bone progressively takes on more load as it heals, while the implant material slowly vanishes. In essence, biodegradable implants aim to provide adequate strength and simultaneous remodeling within the bone during healing [14,15].

We performed a narrative literature review by searching PubMed, Web of Science, and Scopus for studies published using the terms ‘PLGA’, ‘biodegradable implant’, ‘pediatric fracture’, and ‘orthopedic fixation’. Clinical studies, case series, meta-analyses, and narrative reviews reporting outcomes of PLGA-based implants in pediatric trauma were considered. When available, comparative studies between absorbable and metallic implants were prioritized. The search was complemented by examining the reference lists of relevant articles. The review summarizes key findings, with particular emphasis on clinical results, complications, and future directions. Limitations of this approach, such as possible selection bias and heterogeneity of included studies, are acknowledged.

## 2. Clinical Applications and Case Examples

**Forearm Fractures:** One of the most common fracture sites in the pediatric population. Elastic stable intramedullary nailing (ESIN) with metal rods is a standard treatment for unstable diaphyseal forearm fractures. However, nowadays, bioabsorbable IM-nails made of PLGA are an alternative to metal rods. In a cohort from 2024 (PLGA 85:15, Activa IM-nail), all 38 patients achieved bone union with stable alignment [25]. At one-year follow-up, the children showed nearly full recovery of their range of motion (ROM), and minor reductions in forearm rotation and elbow flexion were not clinically significant. No complications, such as refractures or irritation, were reported. A randomized trial showed that absorbable IM nails achieved the same long-term forearm rotation and function as titanium ESIN at ≥4 years, with most implants fully or almost entirely degraded on MRI and no unexpected degradation-related problems; only two early implant failures occurred within three months postoperatively [20]. The meta-analysis by Eastwood et al. pooled 255 children (399 forearm fractures) treated with bioresorbable implants and found that PLGA IM nails achieved union rates and functional outcomes comparable to standard metal ESIN, with a low overall mechanical failure rate (~3%) and no signal of increased serious complications. The literature confirms that RIN is a safe and effective, which produces results that are comparable to traditional metal implants [15,20,26,27].

**Distal Radius Fractures:** Severely displaced distal radius and distal metaphyseal forearm fractures in children have been successfully managed with PLGA intramedullary pins (ActivaPin, 2.0–3.2 mm diameter) as an alternative to percutaneous K-wires. A retrospective multicentric study of 94 children found that PLGA pins achieved equivalent outcomes compared to K-wires. Complications were significantly less, namely, they avoided typical problems such as pin track infections or irritation [28]. At 18-month follow-up with MRI, no growth disturbances were detected in the PLGA group [28]. After 1.5 years of follow-up, there were no growth disturbances observed in any patients, indicating that neither the biodegradable implants nor the K-wires affected the physes negatively [28].

**Ankle (Physeal) Fractures:** Fractures of the distal tibia involving the growth plate (Salter–Harris fractures) are another scenario where implant choice is critical. Metal screws across a growth plate must be removed promptly to avoid growth arrest. A retrospective study of 128 pediatric ankle fractures compared PLGA absorbable screws to standard metallic screws for fixing physeal fractures (mainly Salter-Harris II, III, IV of the distal tibia) [29,30]. The study noted that the PLGA implants achieved comparable fracture stability and healing as metal screws, but without necessitating implant removal [30]. In contrast with studies using PLLA/TMC blends (e.g., Inion OTPS), which also avoid removal, exhibit prolonged degradation (>2–3 years) and distinct elastomeric properties that may be less ideal for simple lag-screw compression compared to PLGA constructs. By utilizing a PLGA 85/15 ratio, sufficient initial strength and complete resorption within 2 years can be guaranteed, avoiding the rapid osteolysis seen with historical PGA implants or the late-onset inflammation associated with crystalline PLLAs [24,29,31,32,33,34].

**Elbow Fractures:** Injuries of the lateral humeral condyles are the second most common elbow fractures in children, which often require operative solutions. Traditionally, management includes K-wires or screws, but as of late, biodegradable pins offer an alternate solution. In lateral condyle fractures, Kassai et al. (2024) used PLGA 85/15 pins (ActivaPin) combined with absorbable sutures as a biodegradable tension band, achieving union rates and alignment equivalent to metal tension bands while eliminating removal surgery [35]. This PLGA-based approach avoids the prolonged degradation and potential osteolysis associated with historical PGA pins or the slower resorption of SR-PLLA used in medial epicondyle fixation [36,37,38]. Similarly in medial epicondylar fractures, it was demonstrated that PLGA pins provide a reliable fixation with no permanent complications, thus offering a more favorable degradation profile (12–24 months) compared to stiffer, longer-lasting PLLA alternatives [5,39,40].

**Other Applications:** Plates were used with success for the fixation of the clavicle [41]. Osteochondral fractures of the patella, as well as femoral condylar fractures, which affect the articular surfaces, were treated with resorbable nails, pins, and screws [42,43,44]. Implants have also been used in other pediatric orthopedic scenarios, including fractures of the radial neck, tibial eminence avulsion fractures, osteotomies for deformity correction, and even spinal deformity surgery in experimental settings [34,45,46].

## 3. Discussion

The main advantage of resorbables over metal implants is that they do not require removal. Avoiding routine implant removal results in lower expenses and a more rapid return to everyday life, as well as decreased risk of complications due to removal [18,36]. The polymers used may invoke only a minimal inflammatory response as they degrade; as reported in the literature, they typically do not elicit significant adverse tissue reactions [22]. Any local acidity from degradation is usually buffered by the body without incident [23,47]. A systematic review reported that no implant-related toxicity or systemic effects were noted, underscoring their safety. As the implant absorbs, it gradually transfers load back to the healing bone in a more natural manner. Progressive loading may stimulate bone remodeling and avoid stress-shielding effects that sometimes occur with stiff metal implants. Continuous remodeling will yield a well-healed bone without the risk of refracture that might occur immediately after rigid metal implants are removed. Additionally, it is particularly useful for percutaneous techniques, as pins avoid having external wires protruding from the skin, decreasing the chance of soft tissue reaction or skin infections, as well as improving comfort during the healing phase [28].

### 3.1. Challenges and Limitations

PLGA implants have lower strength and stiffness than metallic counterparts [14,48]. An elastic modulus of 3–7 GPa vs. 110 GPa and a tensile strength of about 50 MPa restricts use to moderate load fractures, contraindicating applications to injuries of the femoral shaft or patients with a high BMI. Implant breakage or migration occurs in 3–5% of cases, often linked to technical factors such as undersized implants or cortex perforation, rather than inherent material failure. While PLGA implants are strong enough for many pediatric fractures, they are not yet suitable for very large load-bearing applications, for instance, femoral shaft fractures, open fractures, or children with increased BMI, where metal still excels [14,15,17,49]. There is a risk that a bioimplant could fail (bend, break, or lose fixation) if subjected to excessive stress or weight before the bone heals. Furthermore, the physical properties of polymer implants (like their flexibility) can make them tricky to handle during surgery. Additionally, as the implants are not visible on plain radioimaging, placement that requires video confirmation is often challenging and requires experience [6,22,50]. A large observational series with PLGA forearm nails recorded a small number of complications, such as implant breakage or migration, leading to issues like secondary displacement or refracture in a few patients [26,27,51]. Nonetheless, these events highlight that meticulous surgical technique and proper implant selection are critical to avoid early failure. In the rest of the cases, refracture occurred due to repeat trauma to the affected limb, highlighting that the correct choice might include the use of metal implants, for example, in the case of highly active children or in the case of children with disabilities. 

Although generally biocompatible, polymer degradation can lead to localized tissue reactions [47,52,53]. The acidic byproducts of PLGA resorption may cause transient inflammation, fluid accumulation, or a mild foreign-body reaction in the surrounding tissue [14,19,47]. Historically, fast-degrading materials, such as PGA or PLA pins, occasionally produced inflammatory reactions, sterile abscesses or osteolytic areas as they degraded. Modern PLGA implants, with optimized composition, have largely mitigated these issues, but surgeons should be aware of the potential. Patients might experience some localized swelling or discomfort as the implant resorbs, usually self-limited [14,20,50,51,52]. Severe reactions might necessitate a minor intervention (e.g., draining a fluid collection) [20,54,55]. Large-scale studies in recent years have reported no implant-related adverse reactions requiring medical treatment [50,51,52]. Technical challenges include screw head shearing under excessive torque and radiolucency, making radiographic confirmation of placement and integrity difficult [14,47,56,57]. The adoption of the material is slow due to upfront costs and limited availability. Unfortunately, biodegradable implants are not as universally available as standard metal implants. The initial cost per unit is generally higher than a simple stainless-steel screw or K-wire [58,59].However, the savings of eliminating a second surgery potentially amount to more. In some regions, a lack of familiarity can be a barrier to use. Surgeons and operating room staff may require additional training to become familiar with equipment handling.

### 3.2. Future Research and Outlook

The positive clinical experience to date with biodegradable implants sets the stage for broader adoption and further innovation [17]. Most studies so far report follow-up outcomes up to about 1–2 years after fracture fixation, which typically show successful healing and no early complications [6]. Nonetheless, it is suggested to examine long-term outcomes well into skeletal maturity. Data on large patient cohorts may help identify any rare adverse events, while comparing functional outcomes, complication rates, cost-effectiveness, and patient-reported outcomes between bioabsorbable implants and standard care would highlight the benefits of modern implant types. The development of biodegradable metal implants, especially magnesium-based alloys, has also gained attention in recent years [14,15,18,60].

Magnesium (Mg) and Mg-based alloys can effectively dissolve in the body by way of corrosion, though with the advantage of metallic resilience [14,18,61]. Early clinical use of magnesium screws in children has shown good biocompatibility and fracture healing with complete implant resorption over time [62]. Magnesium screws achieved bone union, with no implant-related adverse reactions and no need for removal, as reported by Baldini et al. [5,14]. This favorable performance indicates that magnesium-based implants could address some constraints of polymer appliances. However, during the degradation of Mg-implants, hydrogen gas is released, which may form a pocket of gas around the implant, marked by a radiolucent area on X-ray. The gas produced during degradation could gradually be reabsorbed and poses no harm to surrounding tissues according to some in the literature; however, there have been reports about adverse gas pocket formation. Continued development strives to design alloys and coatings that sufficiently regulate the rate of corrosion and further reduce or balance gas release. These strengthened implants could take on load-bearing applications that current polymers cannot, expanding the scope of resorbable fixation devices [14,62,63].

Fiber-reinforced polymers (such as self-reinforced PLA) or bioactive ceramic particles that are osteogenetic during degradation are under evaluation [48,64]. Complications arising due to initial flexibility or sudden loss of mechanical integrity can be overcome by tailoring the microstructure of the polymers [17]. Drug-eluting bioimplants could further revolutionize orthopedic care [49,65]; implants that could release antibiotics or growth factors locally as they break down, thereby preventing infection or enhancing healing [65,66]. These multi-functional implants could be highly valuable in trauma cases with high infection risk (open fractures) or in scenarios with problematic healing [67]. Future directions are not only about the material but also implant design and how surgeons may utilize them. Computer-aided design and 3D printing are enabling patient-specific biodegradable implants, customized to the child’s anatomy and fracture pattern [68]. Researchers have begun exploring bioresorbable 3D-printed splints and plates that precisely fit complex shapes (acetabular fragments or small bones of the hand) [69,70]. To ensure safety and adequate selection of implant type, oversight is necessary with appropriate guidelines due to the expansion of use. Constant developments in bone and tissue engineering have led to safer, less invasive orthopedic practices with fewer procedures required.

Limitations of this narrative review include the heterogeneity of the findings, with varied study designs, implant types, and follow-up durations, which makes plain comparison challenging. It does not include a systematic literature search or meta-analysis, so the selection of studies may be subject to bias. Moreover, long-term outcomes of biodegradable implants beyond several years are still limited in the literature, so the review cannot provide definitive conclusions on their effects well into skeletal maturity. Finally, because this review does not present new experimental or clinical data, its role is primarily to summarize current trends and applications rather than establish new evidence.

## 4. Conclusions

Biodegradable implants have introduced an important paradigm shift in the management of fractures. By providing stable fracture fixation that progressively loads the bone and resorbs, these implants directly address the longstanding issue of routine hardware removal in children. Clinical experience over the last ten years—spanning forearm, wrist, ankle, and elbow injuries—demonstrates that modern bioresorbable devices (particularly PLGA-based implants) can achieve bone healing and functional outcomes equivalent to those of traditional metal hardware. At the same time, they confer unique benefits: children avoid additional surgeries, and the risk profile of treatment is improved with fewer implant-related complications. The safety record of these materials in growing patients has been reassuring, with high biocompatibility and minimal adverse reactions reported.

## Data Availability

The data presented in this study are available in the article.

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
