# Peer review of "Biodegradable (PLGA) Implants in Pediatric Trauma: A Brief Review"

_children, 2025, doi:10.3390/children13010019_

Round 1

Reviewer 1 Report

Comments and Suggestions for Authors

This manuscript reviews the use of PLGA bone fixator implants in pediatric trauma, but the content is limited and lacks depth, consistent with its self-identified status as a "brief review."

The review's scope is overly generalized. Aside from the initial two paragraphs of the Introduction and parts of Section 2 (Clinical Applications and Case Examples), the manuscript fails to go deeply into the specific context of pediatric trauma

Similarly, the Discussion and Future Perspective sections offer generalized comments on absorbable PLGA bone fixators rather than providing specific, novel insights specificall to padiatrix cases.

Overall, the manuscript's contribution to the field is limited. The knowledge gained is comparable to existing, more systematic and detailed reviews on bioabsorbable bone fixators i.e

1.Lu, Y., Zhang, T., Chen, K., Canavese, F., Huang, C., Yang, H., Shi, J., He, W., Zheng, Y., & Chen, S. (2025). Application of biodegradable implants in pediatric orthopedics: shifting from absorbable polymers to biodegradable metals. Bioactive Materials, 50, 189 - 214. https://doi.org/10.1016/j.bioactmat.2025.04.001.

2.Grün NG, Holweg PL, Donohue N, Klestil T, Weinberg AM. Resorbable implants in pediatric fracture treatment. Innov Surg Sci. 2018 May 29;3(2):119-125. doi: 10.1515/iss-2018-0006. PMID: 31579775; PMCID: PMC6604569.

3.Phong Truong, Kristina Kuklova, Gino Mercadal, Diego Galindo. Resorbable Orthopedic Implants in Pediatric Patients: A Narrative Review. Glob J Ortho Res. 3(3): 2021. GJOR.MS.ID.000561

4.Jozsa, G., & Varga, M. (2025). Pediatric Fractures Treated by Resorbable Implants. IntechOpen. doi: 10.5772/intechopen.1009144

Reviewer 2 Report

Comments and Suggestions for Authors

The authors present a narrative review analyzing the impact of bioabsorbable materials, particularly PLGA, in the fabrication of pediatric orthopedic implants, with a focus on reported trends and clinical applications. The topic is relevant and timely, as bioabsorbable materials represent a promising alternative to conventional metallic fixation devices, potentially improving patient outcomes and reducing the need for secondary removal surgery.

The manuscript provides a useful overview of the subject; however, the literature search could be further expanded and updated, incorporating the most recent evidence published within the last five years. This would strengthen the scientific value of the work and ensure that the review accurately reflects current developments in the field.

The authors correctly highlight the advantages of bioabsorbable implants, such as avoiding a second surgical procedure for hardware removal, reducing surgical complications, and simplifying postoperative pipeline. Nevertheless, a more critical discussion of potential risks and limitations is recommended, considering both the clinical aspects (e.g., biocompatibility, degradation behavior, tissue response) and the technical aspects related to the mechanical performance and manufacturing of such implants.

Specific comments:

  • Line 142: please correct the citation.
  • Expand the number of references concerning the clinical applications discussed and explore possible correlations between implant type and bioabsorbable material used.
  • Update the reference list to include recent studies (within the past 5 years).

Alongside the reported benefits, please discuss clinical and/or technical limitations that may currently restrict or, conversely, further support the adoption of these materials in orthopedic practice.

Reviewer 3 Report

Comments and Suggestions for Authors

Fractures in children can be managed with casting or, when more severe or complex, treated surgically. Typically, fixation devices are inserted to stabilize the fracture and must later be removed. However, this approach presents several disadvantages, including increased hospital management costs, additional visits and procedures required from families, and a higher risk of infections and other complications associated with performing two separate surgeries.

The use of resorbable fixation systems eliminates the need for hardware removal, reducing these burdens. In this manuscript, the authors provide a brief literature review on this topic, covering different anatomical fracture scenarios and focusing on PLGA-based devices to assess their performance.

  • Line 279: The sentence “Clinical experience over the last ten years” would fit better if the authors added a short paragraph after the Introduction (around line 116) describing the search criteria used for selecting the papers included in the review. Even though this is not a systematic review, a brief explanation of the methodology would strengthen the manuscript and help contextualize the summarization of current trends.

Round 2

Reviewer 1 Report

Comments and Suggestions for Authors

The authors have revised the manuscript based on the recommendations provided. However, the revised manuscript is still too generalized and does not sufficiently focus on pediatric applications.   The review already provides insights into the clinical application of PLGA in various pediatric fracture patterns. However, revisions are needed in two main areas as follows:

1. The current discussion of PLGA is too brief. Authors should expand this section to clearly justify why PLGA is the superior choice for the growing skeleton compared to related polymers i.e, PLLA, PGA. Also, its suitability for pediatric applications. 

2. While the clinical outcome sections are relevant, the introduction, background, and general discussion sections often lose focus, treating PLGA as a general product for patients rather than one for children. Please address the following throughout the text:

  • Degradation Time: Explain explicitly that the required lifespan of the implant is dictated by the child's rapid healing rate; the implant must maintain stability until fracture union but be fully gone before it can interfere with growth plate (physis) function.

  • Risk Profile: Frame the discussion of complications (e.g., inflammation, cyst formation) within the context of the open physis. Does the acidic degradation byproduct pose a unique risk to the actively dividing cells of the growth plate? This is the core pediatric distinction that must be addressed.

  • Future Directions: Maintain consistency by focusing the future outlook exclusively on absorbable polymer advancements and how they address pediatric-specific issues. Do not introduce unrelated material like Mg alloys, as this confuses the scope.
